# Time-lapse crystallography snapshots of a double-strand break repair polymerase in action

Joonas A. Jamsen [1], William A. Beard[1], Lars C. Pedersen[1], David D. Shock[1], Andrea F. Moon[1], Juno M. Krahn[1], Katarzyna Bebenek[1], Thomas A. Kunkel[1] & Samuel H. Wilson[1]

DNA polymerase (pol) μ is a DNA-dependent polymerase that incorporates nucleotides during gap-filling synthesis in the non-homologous end-joining pathway of double-strand break repair. Here we report time-lapse X-ray crystallography snapshots of catalytic events during gap-filling DNA synthesis by pol μ. Unique catalytic intermediates and active site conformational changes that underlie catalysis are uncovered, and a transient third (product) metal ion is observed in the product state. The product manganese coordinates phosphate oxygens of the inserted nucleotide and $PP_i$. The product metal is not observed during DNA synthesis in the presence of magnesium. Kinetic analyses indicate that manganese increases the rate constant for deoxynucleoside 5′-triphosphate insertion compared to magnesium. The likely product stabilization role of the manganese product metal in pol μ is discussed. These observations provide insight on structural attributes of this X-family double-strand break repair polymerase that impact its biological function in genome maintenance.

[1] Genome Integrity and Structural Biology Laboratory, National Institute of Environmental Health Sciences, National Institutes of Health, Research Triangle Park, NC 27709, USA. Correspondence and requests for materials should be addressed to S.H.W. (email: wilson5@niehs.nih.gov)

DNA synthesis is a fundamental reaction required for essential processes such as DNA replication and repair. DNA polymerases catalyze incorporation of a deoxynucleoside 5′-triphosphate (dNTP) at the end of a primer DNA strand, extending the primer by one nucleotide (dNMP) and producing pyrophosphate (PP$_i$). Historically, DNA polymerases have been thought to utilize a two-metal mechanism that is shared by other nucleotidyl transferase enzymes[1–3]. However, many important features of the catalytic mechanism of DNA polymerases have remained elusive, in part, due to a lack of high-resolution crystal structures of intermediate states during the catalytic cycle. Recent structural descriptions of nucleotide insertion by DNA polymerases β[4, 5] and η[6, 7] employed time-lapse X-ray crystallography to capture snapshots of catalysis within the crystal and reveal novel intermediates during the catalytic cycle. This approach employs natural substrates, instead of substrate analogs, to follow the reaction in crystallo and provides structures of the liganded enzyme before, during, and after catalysis. Thus, time-lapse X-ray crystallography can reveal conformational adjustments and structural features as a polymerase (pol) transitions from one liganded form to another. In particular, an adjunct third (product) metal was observed to coordinate product phosphate atoms during time-lapse analysis of pol β, along with loss of the essential active site catalytic metal[4, 5, 8]. Although additional metals of unknown function were previously observed near the active sites of DNA and RNA polymerases[9–11], computational studies with pol β suggested that the third (product) metal does not lower the energy barrier for the forward reaction[12], but suppresses the reverse reaction[13]. In the case of pol η, it has been suggested that the third (product) metal promotes the forward reaction[7] or may facilitate product release[14].

Based on sequence homology, the mammalian DNA polymerases are classified into families: A, B, X, Y, and reverse transcriptase[15–17]. The X-family includes pols β, λ, μ, and terminal deoxynucleotidyl transferase (TdT)[18, 19]. These pols are DNA repair enzymes contributing to genome maintenance. X-family members share a common carboxyl-terminal pol domain with significant sequence and structural homology. Single-nucleotide gap filling by pol β, the smallest eukaryotic X-family member, has been extensively characterized both structurally and kinetically[20]. In brief, an "open" pol/DNA binary complex binds the incoming dNTP along with a magnesium ion, referred to as the "nucleotide" metal (Mg$_n$). This binding event triggers formation of the "closed" ternary complex[21]. The "closing" transition repositions the carboxyl-terminal N-subdomain to sandwich the nascent base pair between polymerase side chains and the duplex DNA terminus[22]. Binding of the catalytic magnesium (Mg$_c$) results in formation of the pre-catalytic "ground state" ternary complex, with two metals, poised for chemistry. The catalytic metal ion facilitates deprotonation of the deoxyribose 3′-hydroxyl group of the primer strand[23]. Nucleotidyl transfer through an in-line attack of the 3′-oxyanion at the α-phosphate of the incoming dNTP results in phosphodiester bond formation and breaking of the P$_α$–O$_{αβ}$ bond of the incoming nucleotide, generating PP$_i$. After chemistry, conformational changes occur resulting in polymerase "opening" and release of products.

Pol μ functions in non-homologous end joining (NHEJ) DNA repair, a major double-strand break repair pathway[24]. Among X-family members, pol μ is most closely related to TdT (41% identity between human paralogs) and possesses the ability to bridge non-complementary DNA ends during NHEJ. Structures of pre- and post-catalytic complexes of human[25] and murine[26] pol μ have been described, and these structures informed computational studies[27, 28] to decipher the enzyme's catalytic and fidelity mechanisms[29]. In contrast to pol β, these

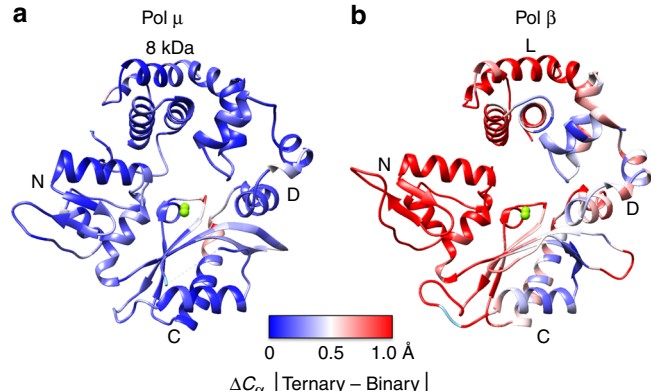

**Fig. 1** Global conformational changes ($\Delta C_\alpha$) upon binary-to-ternary complex transition in (**a**) pol μ and (**b**) pol β. The structures shown are the ternary complexes of each enzyme. The protein backbones are colored as a heat map from *blue* to *red* corresponding to changes in $C_\alpha$ positions on a scale from 0 to 1.0 Å, calculated from an alignment of the binary (PDB ids 4LZG and 3ISM for pol μ and pol β, respectively) and ternary (PDB ids 4M04 and 2FMS for pol μ and pol β, respectively) complexes using the Matchmaker tool in the Chimera package[46]. All the $C_\alpha$ of the respective pol μ structures were used to perform the alignment, as pol μ does not display subdomain motions (RMSD, 0.23 Å). In contrast, since the N-subdomain of pol β undergoes repositioning when a nucleotide binds to form the ternary complex (Fig. 4a in ref. [20]), the alignment was restricted to the other pol β subdomains (RMSD, 0.86 Å, 224 residues). The domains/subdomains are indicated; the polymerase domain includes the D-, C-, and N-subdomains corresponding to the thumb, palm, and fingers of right-handed polymerases[20], as well as the 8-kDa and lyase (L) domain of pol μ and β, respectively. Active site metals are shown as *green spheres* and the ends of the disordered loop region in pol μ are shown connected by a *dashed line* (*cyan*)

studies indicated that pol μ lacks large subdomain motion, as it transitions during the catalytic cycle among various liganded complexes (Fig. 1). Thus, pol μ is rigid, with only limited conformational adjustments as it converts from binary (DNA), ternary substrate (DNA and dNTP) and post-catalytic (DNA$_{n+1}$ and PP$_i$) complexes[25, 26]. Based on in vitro and in vivo functional assays, the enzyme is able to accommodate a wide range of DNA substrates and is able to incorporate both dNTPs and NTPs[30, 31]. Since these early structural studies made use of substrate analogs, we now have employed natural substrates and time-lapse X-ray crystallography to capture molecular events as the pre-catalytic complex is converted to a product complex.

## Results

**Time-lapse X-ray crystallography.** We applied time-lapse crystallography to visualize single-nucleotide insertion by pol μ in the presence of either Mg$^{2+}$ or Mn$^{2+}$ using a single-nucleotide gapped DNA substrate; this substrate had been used previously to structurally characterize pre- and post-catalytic complexes[25, 26, 29]. In addition, a similar gapped DNA was used in time-lapse crystallography with pol β[4, 5], a model X-family polymerase. Initially, a pre-catalytic ground state (GS) ternary substrate complex was formed in crystallo (Fig. 2a). To accomplish this, a crystal of the pol μ/DNA binary complex was obtained in the absence of a divalent metal ion, and then the crystal was soaked in a cryo-solution containing CaCl$_2$ and dNTP for 15 min at 4 °C; note that calcium does not support catalysis. The Ca$^{2+}$-bound GS crystal was transferred to a cryo-solution containing Mg$^{2+}$ or Mn$^{2+}$. The bound Ca$^{2+}$ is replaced by Mg$^{2+}$ or Mn$^{2+}$, which support catalysis, thereby initiating the

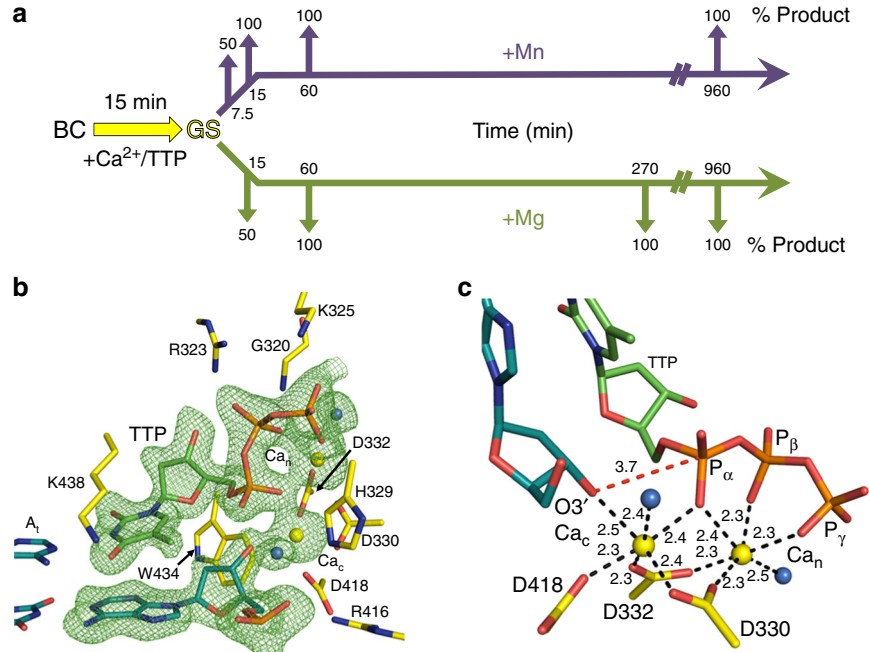

**Fig. 2** Time-lapse crystallography and the structure of the ground state pre-catalytic ternary complex of pol μ. **a** Soaking binary complex (BC, *black*) crystals (gapped DNA/protein) of pol μ in a cryo-solution containing $CaCl_2$ and the incoming nucleotide (TTP) for 15 min results in formation of the $Ca^{2+}$-bound pre-catalytic GS (*yellow*) ternary complex. Subsequent soaking of the GS in a cryo-solution at 4 °C containing 10 mM $MgCl_2$ (*green*) or 10 mM $MnCl_2$ (*purple*) initiates the reaction in the crystal. Crystals are frozen after increasing incubation times (*green* and *purple arrows*) and the structure of the complex is determined. **b** The active site of the $Ca^{2+}$-bound pre-catalytic GS ternary complex. The incoming nucleotide (*green stick*), $Ca^{2+}$ ions (*yellow spheres*), primer terminus and templating adenine base (*cyan stick*), as well as key active site residues (*yellow stick*) are shown. The $F_o$–$F_c$ simulated annealing omit map (*green mesh*) shown is countered at 2.5σ. **c** A 90° rotation of the active site in comparison to **b** with metal-coordinating ligands indicated by *dashes*. Metal coordination is illustrated with *dashed lines* with distances (Å) indicated. The distance between O3′ and Pα is shown as a *red dashed line*

reaction in crystallo. At various times, the reaction was stopped by flash freezing the crystals, and the structures were determined.

**The pre-catalytic GS ternary complex.** A 1.95 Å structure of the pre-catalytic GS ternary complex of pol μ was obtained after the soak in $Ca^{2+}$ and TTP (Supplementary Table 1). The active site is fully occupied with the incoming TTP (opposite templating base adenine) and $Ca^{2+}$ atoms are observed in the catalytic ($Ca_c$) and nucleotide metal ($Ca_n$) binding sites (Fig. 2b). No incorporation of the incoming nucleotide was evident in the GS structure, as revealed by the lack of electron density between O3′and Pα of the incoming nucleotide. Apart from minor coordination distance differences due to the identity of the metal ($Ca^{2+}$ vs. $Mg^{2+}$), the GS structure is very similar to the reported structure of pol μ bound to a non-hydrolyzable incoming nucleotide analog, dUMPNPP and $Mg^{2+}$ (PDB ID 4M04) (Supplementary Fig. 1).

As shown in Fig. 2c, Asp330 and Asp332 coordinate both active site metals. The nucleotide metal coordinates non-bridging oxygens on all three phosphates of TTP. The $Ca^{2+}$ ion in the catalytic metal site also coordinates Asp418, O3′ of the primer terminus, a non-bridging oxygen of Pα (TTP), as well as a water molecule. The electron density, coordination distances (2.3–2.5 Å) and octahedral geometry (Fig. 2c) are consistent with the metal sites being occupied by $Ca^{2+}$. The primer terminus O3′ is 3.7 Å distal to Pα (TTP). Additionally, an interaction is observed with the base of the incoming nucleotide and Lys438 (Fig. 2b). Hydrogen bonds are observed between Arg323 and a non-bridging oxygen of Pβ of the incoming nucleotide, as well as Lys325 and the backbone nitrogen of Gly320 with Pγ. His329 occupies a conformation proximal to Asp330 (conformation hereafter referred to as the "proximal conformation"). In the

presence of $Ca^{2+}$, the Nε2 nitrogen of the histidine ring is hydrogen bonded with a nearby water molecule on the distal side of His329, while in the presence of $Mg^{2+}$, a shift (0.8 Å) in the backbone repositions His329 (Nε2) to coordinate O3B of Pγ and thus stabilize binding of the incoming nucleotide (Supplementary Fig. 1).

**$Mg^{2+}$-mediated insertion.** In order to observe structural changes that accompany the insertion of dNMP into gapped DNA by pol μ, the $Ca^{2+}$ ions occupying the catalytic and nucleotide metal sites were exchanged for a divalent metal that supports catalysis. We soaked crystals of the GS pre-catalytic ternary complex in a cryo-solution containing 10 mM $MgCl_2$ at 4 °C, initiating catalysis in crystallo. After a 15 min soak, the crystals were flash frozen and the structure determined to 2.10 Å (Supplementary Table 1). The structure at this stage of the reaction was very similar to the structure of the GS ternary complex. However, as shown in the $F_o$–$F_c$ simulated annealing omit map (Fig. 3a), density is observed between O3′ and Pα (TTP) and between Pα and Pβ indicating that at this time point some product formation had occurred and that some reactant persisted. Clear inversion at Pα can be observed, indicative of bond formation; occupancy refinement of both the incorporated and incoming nucleotides indicates ~50–60% incorporation of TTP occurred. This is defined as the reaction state (RS). The product $PP_i$, with an occupancy of ~50–60%, is located in a position identical to that expected directly after bond cleavage. Subtle changes in active site side-chain conformations are observed, including the observation that a proportion of His329 has flipped ~150° into a distal conformation and now lies ~5 Å away from Pγ (Fig. 4a). The occupancy of the distal conformation of His329 corresponds to

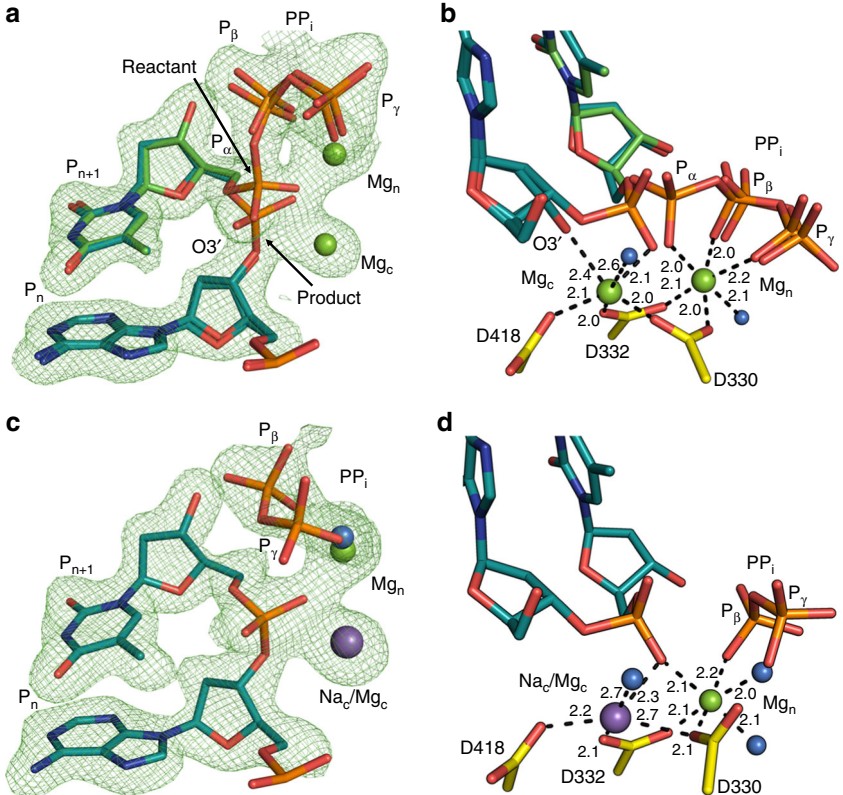

**Fig. 3** Magnesium-mediated insertion. Soaking of the $Ca^{2+}$-bound pre-catalytic GS ternary complex in a cryo-solution containing 10 mM $MgCl_2$ initiates the reaction in the crystal. Crystals were frozen after incubation for the indicated times in the cryo-solution. **a**, **b** Reaction state (RS) at 15 min, **c**, **d** product state (PS) at 60 min. The panels on the *left* (**a**, **c**) display density for the incoming and incorporated nucleotides. The panels on the *right* (**b**, **d**) are close-up views of a 90° rotation of the active site with coordinating ligands indicated by *dashes* and labeled distances (Å). The incoming TTP and incorporated nucleotides (TMP) are in *cyan* and *green stick* representation, respectively, while the protein side chains are in *yellow*. $Mg^{2+}$ ions are displayed as *green spheres* and water molecules as *blue spheres*. Sodium is shown as a large *purple sphere*. The $F_o$–$F_c$ simulated annealing omit map (*green mesh*) shown is contoured at 2.5σ

the amount of $PP_i$ formed and dNMP incorporated (~50–60%). Bond lengths, angles, and coordination geometry are consistent with occupancy of the catalytic and nucleotide metal-binding sites with $Mg^{2+}$, along with some $Ca^{2+}$ (~50%) in the catalytic metal-binding site due to incomplete exchange with $Mg^{2+}$. Although a third or product metal was seen to coordinate the product phosphates, $DNA_{+1}$ and $PP_i$, in the product state of pol η[6], as well as of pol β[4, 5, 8], a product metal is not observed in this structure.

In order to examine if a product metal would appear at a higher $Mg^{2+}$ concentration, we performed a 15 min soak in a cryo-solution containing 100 mM $MgCl_2$. The resulting 1.65 Å structure displayed the same degree of turnover as the soak at the lower $MgCl_2$ concentration and lacked density for the product metal. Additionally, structures at earlier and later time points (~30 and 60% turnover) were identical to the reaction state structure, apart from increasing occupancies of the incorporated nucleotide, product $PP_i$ and the distal conformation of His329, as the reaction progressed (Supplementary Table 1). Concomitant decreases in the occupancy of the incoming nucleotide were observed with time of reaction. Importantly, density was not observed in the location of the product metal site in any of the above structures.

Following a 60 min soak, we determined the structure of the product complex (PS) of pol μ to 2.02 Å (Supplementary Table 1). The $F_o$–$F_c$ simulated annealing omit map in Fig. 3c indicates full bond formation between O3′ and Pα (TTP), bond breakage between Pα and Pβ, and concomitant accumulation of $PP_i$. The

position of Pβ of the $PP_i$ product is identical to that expected after bond cleavage, but the oxygen ligand of Pγ is more disordered, with a water replacing the interaction with $Mg_n$. Thus, density remains for Pβ, suggesting that Pγ dissociates prior to Pβ to facilitate product release (Fig. 3c). His329 flips ~150° into the distal conformation and Asp330 rotates ~90° to adopt a post-catalytic conformation (Fig. 4a). Based on coordination distances and geometry, the nucleotide metal-binding site is fully occupied by $Mg^{2+}$ coordinating the $PP_i$ leaving group oxygens, Asp330 and Asp332, as well as a water molecule. The atom in the catalytic metal-binding site, however, now possesses longer coordination distances (~2.1–2.7 Å) with loss of the O3′ coordinating ligand, consistent with $Na^+$ at this site. Sodium has therefore partially replaced $Mg^{2+}$ in the catalytic site, in a fashion similar to that at the corresponding site during the pol β catalytic cycle[2].

To observe post-catalytic events in the pol μ reaction cycle (e.g., $PP_i$ and metal release), we performed extended soaks, up to 960 min, of the product complex. The structures were generally similar to the product structure after the 60 min soak. A 1.97 Å structure was obtained after a 270 min soak and displayed full turnover (Supplementary Fig. 2a, b). The nucleotide metal site is occupied by $Mg^{2+}$ and the catalytic site is now fully occupied with $Na^+$. A 960 min soak structure (PS*) was determined at 1.73 Å, and the structure is shown in Supplementary Fig. 3. Based on coordination distances and geometry, the nucleotide metal-binding site is occupied by $Mg^{2+}$, while $Na^+$ still occupies the catalytic metal site. His329 is in the distal conformation, while

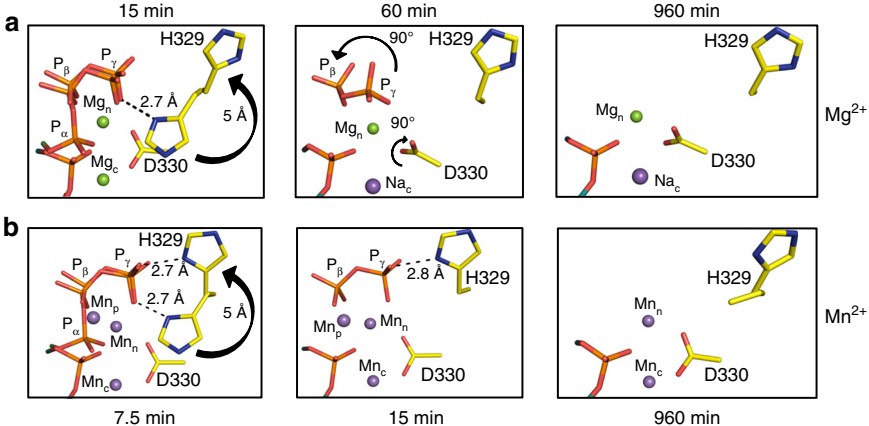

**Fig. 4** Active site conformational adjustments during single-nucleotide gap-filling by pol μ. **a** $Mg^{2+}$-mediated insertion, and **b** $Mn^{2+}$-mediated insertion. *Black arrows* indicate movement compared to the previous panel from *left* to *right*

Asp330 is in the post-catalytic state in both the 270 and 960 min soaks (Fig. 4a). Importantly, both extended soak product structures suggest $PP_i$ release. The density present at the $PP_i$-binding site is consistent with a potential cryo-solution contaminant, glycolate, that was modeled into the active site. However, the presence of a small amount of $PP_i$ bound to the active site cannot be excluded.

**$Mn^{2+}$-mediated insertion**. Based on enzymatic assays, manganese has been suggested to be the physiological metal co-factor of pols λ and μ[32–34]. Since the electron density of a magnesium atom with partial occupancy cannot be readily distinguished from a water molecule, the anomalous signal produced by $Mn^{2+}$ was employed to identify active site metals. Accordingly, structures determined along the reaction pathway of $Mn^{2+}$-mediated catalysis by pol μ were obtained. A structure of pol μ at 7.5 min after reaction initiation was solved at 1.93 Å (Supplementary Table 1). The structure at this stage of catalysis displays ~60% turnover (Fig. 5a) and is very similar to the structure of the $Mg^{2+}$-catalyzed reaction, where density for a mixture of reactant and product species was observed. In this case, however, a third (product) $Mn^{2+}$ is observed coordinating Pα of the incorporated nucleotide and Pβ of the $PP_i$ product. Four water molecules complete the octahedral coordinating metal sphere. This site possesses density attributed to $Mn^{2+}$-mediated anomalous dispersion (Fig. 6a), and its occupancy and those of the surrounding water molecules correspond to the percentage of product formation. As in the $Mg^{2+}$-mediated reaction, His329 is seen to flip into a distal conformation ~5 Å away from Pγ, while its occupancy (~60%) corresponds to the amount of $PP_i$ formed. The conformation of His329 in the manganese structure is intermediate between the proximal and distal conformations, where Nε2 is within hydrogen-bonding distance of Pγ (Fig. 4b). Increased dynamics in the $PP_i$ is observed, as positive density near Pγ and the product metal site remained after occupancy refinement of the product $PP_i$ and product metal regions of this structure. The positive density is weak and was not modeled. The catalytic and nucleotide metal-binding sites contain octahedrally coordinated $Mn^{2+}$ with characteristic coordination distances[35]. Structures obtained at earlier (4 min) and later (10 min) time points corresponded to ~40 and ~70% turnover, respectively (Supplementary Table 1). These structures were identical to the 7.5-min reaction state structure, except for increasing occupancies of the incorporated nucleotide, product $PP_i$, the product metal ($Mn_p$), and the corresponding distal conformation of His329, as the reaction progressed. Changes in the conformation of Asp330

were not observed in these structures and this residue stays in the pre-catalytic conformation throughout the $Mn^{2+}$-mediated catalytic cycle (Fig. 4b).

Figure 5c, d illustrates the active site of the product state in the $Mn^{2+}$-mediated reaction 15 min after initiating the reaction. Occupancy refinement indicates 100% occupancy of incorporated TMP with no density for a bond between Pα and Pβ in the $F_o$–$F_c$ simulated annealing omit map (Fig. 5c). His329 displays a distal conformation, where Nε2 is seen to coordinate Pγ. Increased dynamics in $PP_i$ is again evident due to increased positive density near Pγ. Asp330 has not rotated, consistent with the catalytic and nucleotide metal-binding sites being fully occupied by $Mn^{2+}$. The product metal is observed with an occupancy of ~60–70%, coordinating the phosphate of the incorporated nucleotide, Pβ of $PP_i$, as well as four water molecules, as observed in the reaction state described above for the $Mn^{2+}$-partial catalysis complex.

In order to observe any post-catalytic changes associated with the $Mn^{2+}$-mediated reaction, we performed longer soaks (60 and 960 min). A 1.95 Å structure was determined following a 60 min soak (Supplementary Table 1) and is consistent with a product structure with full incorporation of TTP as shown in the $F_o$–$F_c$ simulated annealing omit map (Supplementary Fig. 2c, d). The third (product) metal is still observed in this structure, but at a reduced (40%) occupancy, indicating the product metal has partially dissociated from the active site with the extended soak. The occupancy of $PP_i$ is 100% in this structure. The structure after a 960 min soak (PS*) also is consistent with a product structure. Surprisingly, both nucleotide and catalytic metal-binding sites still possess an anomalous signal (Fig. 6c), and the coordination distances/geometry are consistent with $Mn^{2+}$, in contrast to observations during $Mg^{2+}$-mediated catalysis, where the catalytic metal is seen to dissociate upon product formation. The arrangement of coordinating atoms is consistent with penta-coordinated stabilization of the $Mn^{2+}$ ion in the catalytic metal site. The extended soak is also consistent with $PP_i$ release, where the remaining density can again be modeled as a glycolate (Supplementary Fig. 4). The product metal site is devoid of density at this stage, indicating that the third (product) metal has fully dissociated. Using the anomalous signal conferred by $Mn^{2+}$ as an indicator, we observed the product metal in structures beginning from 40% product formation until long after the reaction is complete.

**Pol μ single turnover kinetics**. Catalysis in crystallo increases with time, as expected for an enzymatic reaction, but is considerably slower than the steady-state reaction in solution at

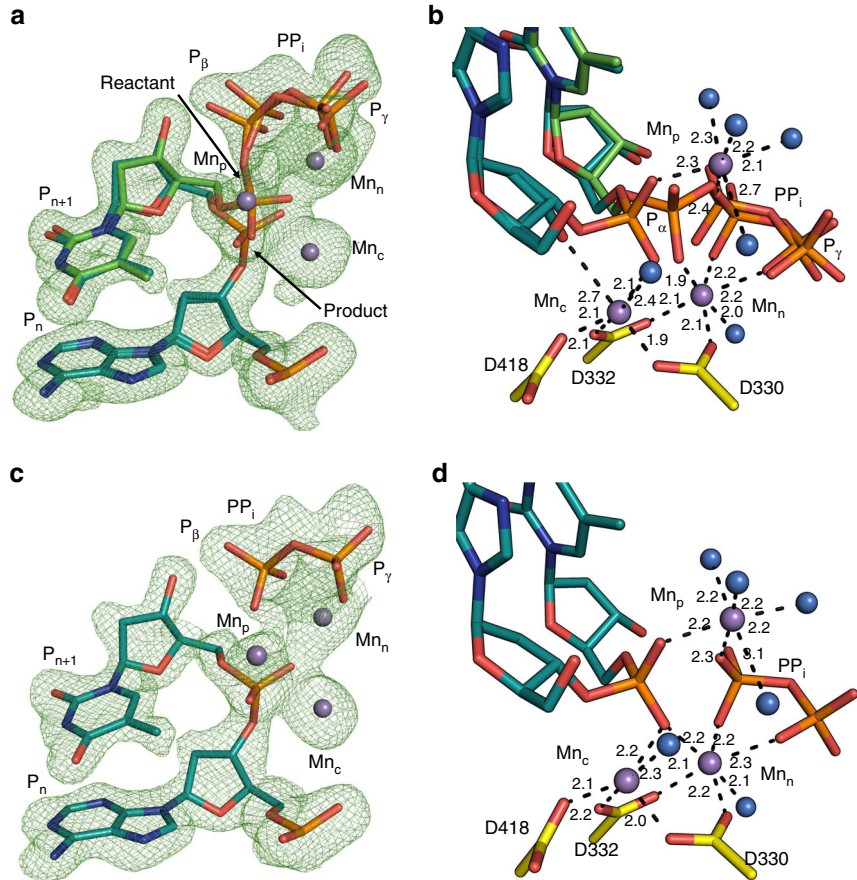

**Fig. 5** Manganese-mediated insertion. Soaking of the $Ca^{2+}$-bound pre-catalytic GS ternary complex in a cryo-solution containing 10 mM $MnCl_2$ initiates the reaction in the crystal. Crystals were frozen after incubation for the indicated times in the cryo-solution. **a**, **b** Reaction state (RS) at 7.5 min, **c**, **d** product state (PS) at 15 min. The panels on the *left* (**a**, **c**) display density for the bond, indicating the degree of completion of the reaction. The panels on the *right* (**b**, **d**) are close-up views of a 90° rotation of the active site with coordinating ligands indicated by *dashes* and distances (Å) labeled. The incoming and incorporated nucleotides are in *cyan* and *green stick* representation, respectively, while the protein side chains are in *yellow*. $Mn^{2+}$ ions are displayed as *purple spheres* and water molecules as *blue spheres*. The $F_o$–$F_c$ omit map (*green*) shown is countered at 2.5σ

physiological temperatures. The decreased activity in crystallo is most likely due to the lower temperature (4 °C) employed during the soaks, presence of inhibitory $Ca^{2+}$ ions, pre-requisite ion-exchange events in a viscous cryo-solution, and other impediments that might be imposed by the crystalline nature of the reaction mixture.

To determine the pol μ nucleotidyl transferase rate, a single-turnover kinetic analysis (enzyme»DNA) of pol μ was performed with $Mg^{2+}$ or $Mn^{2+}$ as the metal ion co-factor. The single-turnover rate constant with single-nucleotide gapped DNA was significantly greater than that reported with a non-gapped substrate[36]. The observed rate constants for the exponential time courses (Fig. 7a) increased hyperbolically with TTP concentration (Fig. 7b), providing estimates of $k_{pol}$ and $K_{d,TTP}$ (Table 1). While the rate of insertion was similar to that expected for a DNA repair polymerase, the apparent binding affinity for the correct nucleotide was much greater (~190 μM) than the physiological nucleotide concentration. Substituting manganese for magnesium only modestly increased the apparent binding affinity. However, $k_{pol}$ for nucleotide insertion increased appreciably, such that overall insertion efficiency increased 50-fold relative to magnesium (Table 1).

Nucleotide analogs with sulfur replacing a non-bridging oxygen on Pα of the incoming nucleotide have been used to analyze whether a chemical step is rate limiting during single-turnover kinetic analysis[37]. Since sulfur is less electronegative

than oxygen, a decreased rate of insertion is expected with sulfur substitution if chemistry is the rate-limiting step. The measured elemental effect for magnesium-dependent nucleotide insertion (Table 1) indicates a strong thio-effect (24-fold). Interestingly, substitution with manganese suppressed the thio-effect, suggesting a change in rate-limiting step from chemistry in the magnesium-dependent insertion, to a non-chemical step in the manganese-dependent insertion.

## Discussion

In this study, we applied time-lapse crystallography to characterize nucleotide insertion by pol μ. Previous time-lapse crystallographic studies of pols β and η uncovered a new active site metal (i.e., third or product metal) thought to have a catalytic role due to its coordination with products[4, 6, 7]. Here we identified the presence of a product metal in the insertion reaction of pol μ. In the presence of $Mn^{2+}$, the product metal site was observed from an early stage (~40%) of the reaction until its completion and beyond. The occupancy of the product metal site correlated with the degree of nucleotide incorporation. After full turnover, the product metal dissociated from the active site, leaving $PP_i$ behind. The $PP_i$ then dissociated from the active site.

The active site of pol μ is similar to the other X-family members, especially that of pol β (Fig. 8). Six conserved active site residues in pols μ and β, including the metal-coordinating

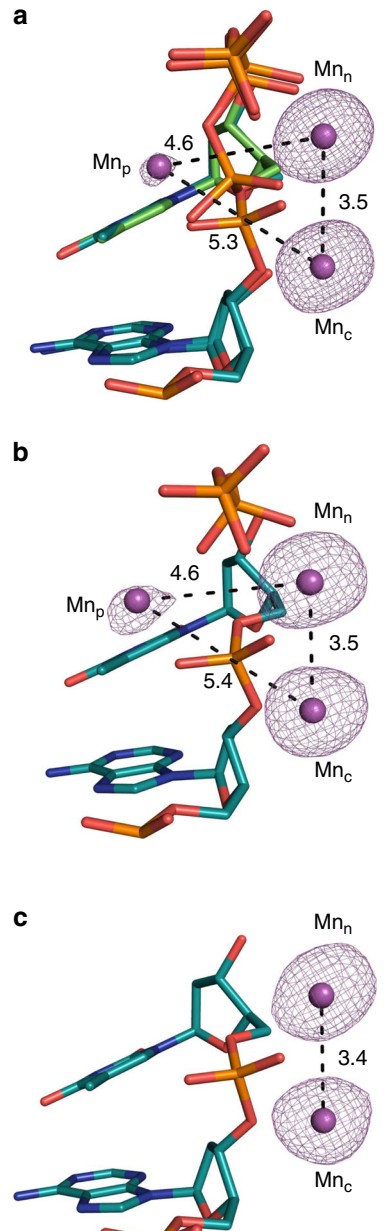

**Fig. 6** Anomalous density map overlayed on structures of the (**a**) RS, (**b**) PS, and (**c**) PS* generated by $Mn^{2+}$-mediated anomalous dispersion. PS* refers to an extended soak (960 min) showing the product metal and $PP_i$ have dissociated, but the catalytic and nucleotide metals remain bound. Anomalous density is shown as a *pink mesh* at 5σ, while the $Mn^{2+}$ atoms are displayed as *magenta spheres*. The incoming nucleotide is shown in *green stick* representation, while DNA is shown in *cyan*. Distances (Å) between metals are labeled and indicated by *dashes*

contrast to observations with pol β, an adjunct third (product) $Mg^{2+}$ is not observed to coordinate products (i.e., phosphate oxygens on the inserted nucleotide and $PP_i$). However, similar to pol β, $Mn^{2+}$ is observed to coordinate product oxygens. These observations suggest that a third metal is not essential for DNA synthesis[12].

With $Mg^{2+}$-dependent insertion, His329 flips ~150° into a distal conformation, the catalytic $Mg^{2+}$ dissociates and is replaced by $Na^+$ (Fig. 4a). Additionally, Asp330 rotates ~90°. In the presence of $Mn^{2+}$, the flip of His329 is observed, but Asp330 remains in the pre-catalytic conformation and maintains its coordination with the catalytic $Mn^{2+}$ throughout the reaction (Fig. 4b). After the $Mg^{2+}$-dependent reaction is complete, $PP_i$ dissociates and is replaced by water molecules (Fig. 4a, *right panel*). Surprisingly, and unlike previous observations, the nucleotide $Mg^{2+}$ remains bound to the enzyme after departure of $PP_i$ indicating that $PP_i$ and nucleotide $Mg^{2+}$ release are uncoupled.

Ligand binding to pol μ is not accompanied by large conformational changes in protein or substrates (Fig. 1a and Supplementary Fig. 5). However, as noted above, differences in side-chain positions do occur in response to metal binding and dissociation. The three active site aspartate residues, required for metal binding and catalysis, do not undergo significant changes in pol β, whereas in pol μ, Asp330 rotates 90° when the catalytic magnesium is released near the end of the reaction. Prior to the rotation of Asp330, His329 shifts from stabilizing Pγ of the incoming nucleotide to the distal conformation, flipping ~150° and repositioning Nε2 ~5 Å away from its initial position (Fig. 4a). This appears to facilitate product release. Asp330 blocks the rotation of His329 into the proximal conformation, and the flip of His329 allows rotation of Asp330 during the $Mg^{2+}$-catalyzed reaction. This rotation of Asp330 is not observed in the presence of $Mn^{2+}$, as the catalytic metal site remains occupied by manganese (Fig. 4b, *right panel*).

Manganese is suggested to be the in vivo metal co-factor of pol μ, as this enzyme is activated in vitro at physiological $Mn^{2+}$ concentrations (i.e., 100 μM) and inhibited at physiological $Mg^{2+}$ concentrations (1–2 mM)[32, 34]. We observe metal rearrangements and associated metal-dependent structural transitions in the pol μ active site during catalysis. The dynamics of the catalytic and nucleotide metals in pol μ follow a similar pattern to those in pol β. Differences in metal function must exist, however, in light of pol μ's increased $Mn^{2+}$-dependent rate constant of insertion, relative to that with $Mg^{2+}$. A similar rate increase for $Mn^{2+}$ over $Mg^{2+}$ is not seen with pol β[32].

Both catalytic and nucleotide metal sites are occupied from the GS to the initial PS with both $Mg^{2+}$- and $Mn^{2+}$-mediated reactions in crystallo. Yet, the exchange of $Mg^{2+}$ with $Na^+$ in the catalytic metal site in the $Mg^{2+}$-catalyzed reaction follows the loss of coordination with O3′, due to bond formation, and Asp330. In contrast, the catalytic metal site remains occupied by $Mn^{2+}$ to the end of the reaction. The stability of $Mn_c$ appears to preclude rotation of Asp330. The nucleotide metal site binds the nucleotide metal as soon as the nucleotide enters the active site, and stability of the nucleotide metal-binding site is indicated by low B-factors in all ternary complex structures, as well as the persistent and strong $Mn^{2+}$ anomalous signal.

In time-lapse crystallography with pol μ in the presence of $Mn^{2+}$, the $Mn_p$ was observed in a similar location as the product metal of pol β (Fig. 8). In pol β, the product metal is surrounded by four coordinating waters, while in pol μ three coordinating waters surround the $Mn^{2+}$ (~2.2–2.4 Å) with a 4th water ~2.7–3.1 Å away (Fig. 5b, d). The product metal site in pol μ is thus less likely to bind $Mg^{2+}$ due to its strict octahedral coordination requirement. In pol μ, the rate of nucleotide

aspartates, overlay closely (Fig. 8a) suggesting similar catalytic roles (Table 2). However, differences between these active sites are found (Fig. 8b) including coordinating residues near Pβ and Pγ of the incoming dNTP (i.e., for pol β/pol μ: Arg149/Lys325 and Gly189/His329) and base and sugar interactions (i.e., for pol β/pol μ: Asp276/Lys438 and Phe272/Trp434). In pol μ, the catalytic metal is positioned to mediate deprotonation of the primer terminus 3′-OH with proton transfer to Asp418 (analogous to Asp256 of pol β, Table 2). Since a water molecule is not apparent near O3′ in the reaction state structure of pol μ (Figs. 3 and 5, panels a, b), proton transfer to water is unlikely. In

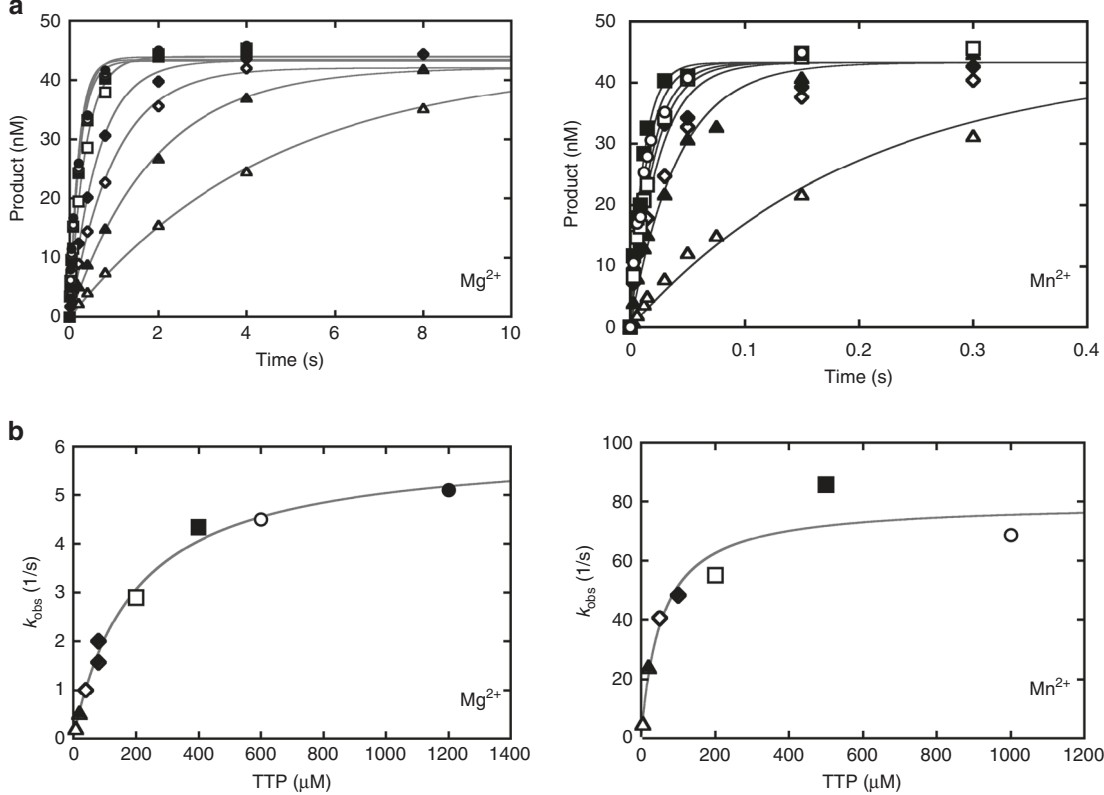

**Fig. 7** Single-turnover kinetics of pol μ. **a** Representative time courses of Mg$^{2+}$- (*left panel*) and Mn$^{2+}$ (*right panel*)-mediated gap-filling insertion of TTP opposite templating dA. **b** Secondary plot of observed single-exponential rate constants against concentration of incoming TTP (Mg$^{2+}$, *left panel*; Mn$^{2+}$, *right panel*). The data were fit to a hyperbolic equation to yield the values listed in Table 1

insertion in the presence of Mn$^{2+}$ in solution (Table 1) and in crystallo (Supplementary Fig. 2a) is higher compared to that observed with Mg$^{2+}$, whereas for pol β, the rate is not dependent on the identity of these two metals[38]. Unlike pol β, catalysis in crystallo is observed in the absence of a product Mg$^{2+}$ for pol μ, indicating that Mg$_P$ is either very transient, precluding its observation, or not essential for catalysis.

Recent computational studies of pol β have shown that Mg$^{2+}$ occupancy of the product metal site does not affect the reaction barrier for the forward nucleotidyl transfer reaction[12], but blocks the reverse reaction[13]. On the other hand, the presence of the catalytic Mg$^{2+}$ was required for the pol β reverse reaction to occur. In pol μ, by analogy to pol β, loss of the catalytic metal in the Mg$^{2+}$-dependent reaction might be expected to block the reverse reaction. In the Mn$^{2+}$-dependent reaction, the retention of Mn$_c$ would be expected to enable the reverse reaction. Nevertheless, in this situation, the presence of Mn$_P$ would be expected to block the reverse reaction[13] and stabilize the PP$_i$ product, thereby promoting the forward reaction.

A significant thio-effect for insertion of the $S_P$-isomer of TTP (αS) in the presence of magnesium provides evidence that chemistry is rate limiting during nucleotide insertion. Historically, a modest thio-effect for polymerase-dependent nucleotide insertion suggested that a non-chemical change limits nucleotide insertion rather than chemistry[37]. Since the theoretical magnitude of the thio-effect on chemistry is uncertain, definitive interpretations concerning modest elemental effects are precluded. For pol μ, however, the strong thio-effect for insertion of the $S_P$-isomer of TTP(αS) in the presence of magnesium indicates that chemistry is rate limiting during nucleotide insertion. Since a strong thio-effect (~24) was observed for pol μ with magnesium (Table 1), the loss of this thio-effect with

manganese is consistent with the observed increase in the rate of manganese-dependent nucleotide insertion, indicating that as the rate of the chemical step is increased (i.e., in the presence of manganese), a slower non-chemical step becomes rate limiting.

While it may be tempting to correlate the presence of a third or product metal with rate limiting chemistry events, two lines of evidence suggest the product-associated metal is not directly involved in chemistry. A critical aspect of this conclusion arises from the fact that the product-associated metal is not observed until after insertion has occurred (i.e., post-chemistry). Second, it is generally accepted that the strong preference for the $S_P$-isomer of the incoming nucleotide is due to its metal-binding characteristics. Accordingly, polymerase insertion studies employing a mixture of $R_P$- and $S_P$-isomers revealed that only the $S_P$-isomer was inserted. Structural studies have confirmed that the pro-$R_P$-oxygen coordinates active site metals while a water molecule coordinates the pro-$S_P$-oxygen in X- and some Y-family DNA polymerases[39]. Importantly, since the pro-$S_P$-oxygen coordinates the product manganese, sulfur substitution at this oxygen would deter binding of the product-associated manganese. This is because manganese is considered thio-phobic preferring to coordinate oxygen[40]. Additionally, the longer P–S bond and larger van der Waals radius of sulfur would preclude product manganese binding[12]. Thus, if the product-associated manganese was directly involved in rate limiting chemistry events, the thio-effect should be much larger than the weak effect observed here.

Pyrophosphate undergoes conformational dynamics during the pol μ catalytic cycle. The PP$_i$ product is observed to rotate ~90° in the 60 min soak with Mg$^{2+}$ (Fig. 3c, d). The latter conformation is stabilized by interactions with Arg320 and O3′ of the newly formed primer terminus. Increased dynamics in Pγ of the PP$_i$

**Table 1 Summary of single turnover kinetic parameters**

| Metal | TTP (αX)[a] | $k_{pol}$ (s$^{-1}$) | $K_d$ (μM) | $k_{pol}/K_d$ (μM-s)$^{-1}$ | $k_{pol(O)}/k_{pol(S)}$ |
|---|---|---|---|---|---|
| Mg$^{2+}$ | O | 6.0 (0.2)[b] | 192 (21) | 0.031 (0.004) | |
| | S | 0.25 (0.01) | 10.5 (1.7) | 0.024 (0.004) | 24 |
| Mn$^{2+}$ | O | 81 (7) | 54 (18) | 1.5 (0.5) | |
| | S | 26 (1) | 27 (5) | 1.0 (0.2) | 3 |

[a]Identity of the atom at the pro-$S_p$ position of Pα (TTP)
[b]Number in parentheses refers to the SE derived from a fit of dNTP dependencies on the observed single-exponential rates to a hyperbolic equation

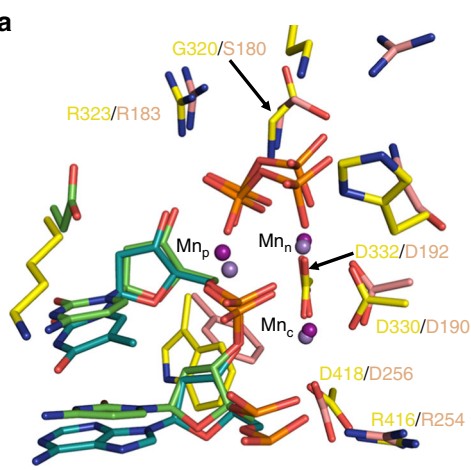

**a**

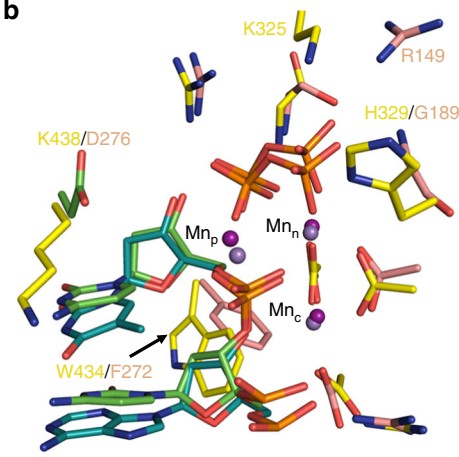

**b**

**Fig. 8** Comparison of the active sites of the pol μ and β product complexes. **a** Structurally equivalent residues: Arg323 (R323), Arg183 (R183); Gly320 (G320), Ser180 (S180); Asp332 (D332), Asp192 (D192); Asp330 (D330), Asp190 (D190); Asp418 (D418), Asp256 (D256); Arg416 (R416), Arg254 (R254). **b** Structurally distinct residues with possible overlapping roles: Lys438 (K438), Asp276 (D276); Lys325 (K325), Arg149 (R149); His329 (H329), Gly189 (G189); Trp434 (W434), Phe272 (F272). Pol μ is displayed in *yellow stick*, while pol β is in *salmon stick* representation. Structurally equivalent manganese atoms are shown in *magenta* and *purple* for pol β and pol μ, respectively. The DNA primer terminus is shown in *cyan* for pol μ and *green* for pol β

product is observed throughout the Mn$^{2+}$-mediated reaction (Fig. 5a, c). This most likely represents conformational flexibility of PP$_i$ that could not be accurately modeled. These additional conformations were therefore omitted from the models. Pγ dissociates first during PP$_i$ release from the pol μ active site, with

**Table 2 Comparison of structurally equivalent residues involved in catalysis in pols β and μ**

| Pol β | Function | Pol μ | Function |
|---|---|---|---|
| Ser180 | Binds β/γ phosphate | Gly320 | Binds β phosphate |
| Arg183 | Binds β phosphate | Arg323 | Binds β phosphate |
| Arg149 | Binds γ phosphate | Lys325 | Binds γ phosphate |
| Gly189 | Binds γ phosphate | His329 | Binds γ phosphate |
| Asp190 | Metal coordination | Asp330 | Metal coordination |
| Asp192 | Metal coordination | Asp332 | Metal coordination |
| Arg254 | Stabilizes Asp256 and PT[a] | Arg416 | Stabilizes Asp418 and PT |
| Asp256 | Coordinates PT and Me$_c$[b] | Asp418 | Coordinates PT and Me$_c$ |
| Phe272 | Modulates Arg258/Asp192 | Trp434 | Stabilizes sugar of PT |
| Asp276 | Stabilizes NT[c] base | Lys438 | Stabilizes NT base |

[a]PT Primer terminus
[b]Me$_c$ Catalytic metal
[c]NT Incoming nucleotide

retention of the Mg$^{2+}$ nucleotide-binding metal (Supplementary Fig. 3). Density for Pγ progressively weakens during the reaction and is almost completely lost in the 270 and 960 min soaks with Mg$^{2+}$ (Fig. 3 and Supplementary Figs. 2a and 3a). Manganese promotes product retention in the active site, as PP$_i$ is still present in the 60 min soak (Supplementary Fig. 2c, d). In the 960 min soak (Supplementary Fig. 4), a glycolate molecule (potential cryo-contaminant) is modeled into density corresponding to the PP$_i$-binding site[41]. The presence of PP$_i$ at this site cannot be excluded, however, the shifted location of the observed density suggests the presence of a non-canonical ligand or PP$_i$ conformation.

The lack of large-scale conformational changes in pol μ or its substrates enables its role in NHEJ. Large-scale conformational changes could impede its biological functions. A rigid platform for synthesis could support pol μ to more effectively utilize variable DNA structures required in repair, since NHEJ processes DNA structures that have dynamic DNA ends (i.e., non-complementary) or DNA strands (i.e., non-contiguous). Curiously, the relatively weak-binding affinity for dNTPs suggests that pol μ could be limited by nucleotide pool concentrations or utilize NTPs that are often found at higher cellular concentrations. Interestingly, a role of the product metal in retarding the reverse reaction could facilitate downstream reactions or interactions with enzymes such as DNA ligase IV.

## Methods

**Protein expression and purification.** Truncated human wild-type pol μ was overexpressed from the pGEXM vector in BL21(DE3)CodonPlus-RIPL cells overnight at 16 °C and purified as follows[25]. Briefly, cells were lysed by sonication in lysis buffer (25 mM Tris, pH 8 (25 °C), 500 mM NaCl, 5% glycerol, 1 mM DTT) and purified in batch on glutathione Sepharose 4B resin (GE Healthcare). Pol μ was obtained by on-resin TEV cleavage overnight at 4 °C. Pol μ was then purified by size-exclusion chromatography on Superdex 200 26/60 in gel filtration buffer (25 mM Tris, pH 8 (25 °C), 100 mM NaCl, 5% glycerol, 1 mM DTT, 1 mM EDTA).

EDTA was added to the gel filtration buffer to remove contaminating divalent metals. The combined fractions (20–30 ml) were then dialyzed against the final buffer (25 mM Tris, pH 8.0 (25 °C), 100 mM NaCl, 5% glycerol, 1 mM DTT), concentrated to ~11 mg ml$^{-1}$, and flash frozen in liquid nitrogen.

**DNA preparation.** Purified oligonucleotides were from Integrated DNA Technologies (Coralville, IA). Short oligonucleotides (i.e., 4-mers) were desalted and used without further purification. A 9-mer template oligonucleotide (5′-CGGCATACG-3′) was annealed with a 4-mer upstream (5′-CGTA-3′) oligonucleotide and a 5′-phosphorylated downstream 4-mer (5′-pGCCG-3′) oligonucleotide in a 1:1:1 ratio to create a duplex DNA with a single-nucleotide gap. Oligonucleotides were dissolved in 100 mM Tris-HCl, pH 7.5, heated to 95 °C for 5 min, and then cooled to 4 °C at a rate of 1 °C min$^{-1}$ and kept on ice until use.

For kinetic assays, a 29-mer template oligonucleotide (3′-CAGTCTGACT GCATACGGCCTGCTGCCTC-5′) was annealed with 14-mer upstream (5′-[Cy3]-GTCAGACTGACGTA-3′) and 14-mer 5′-phosphorylated downstream (5′-pGCCGGACGACGGAG-3′) oligonucleotides to create a single-nucleotide gap (template nucleotide underlined). Oligonucleotides were resuspended in 10 mM Tris-HCl, pH 7.5, 1 mM EDTA, and mixed in a 1.2:1.2:1 ratio (template: downstream:upstream), then annealed as described above.

**Time-lapse crystallography.** Binary complex crystals of pol μ with a templating adenine in a 1-nucleotide gapped DNA were grown at 4 °C, by mixing pol μ (11 mg ml$^{-1}$) with mother liquor (100 mM HEPES, pH 7.5, 16–18% PEG 4000), using the sitting-drop vapor-diffusion technique[25]. Time-lapse crystallography was performed as follows: pol μ–DNA binary complex crystals were transferred to a drop consisting of a cryo-solution at 4 °C containing: 15% ethylene glycol, 100 mM HEPES, pH 7.5, 20% PEG4000, 5% glycerol, 50 mM NaCl, 1 mM TTP, and 10 mM CaCl$_2$ for 15 min. The GS ternary complex crystals formed in this manner were then transferred to the same cryo-solution preceded by a pre-soak wash (to remove excess CaCl$_2$), but instead of CaCl$_2$, containing 10 mM MgCl$_2$ or 10 mM MnCl$_2$ for varying times, and lacking the incoming nucleotide. The process was terminated by plunging the crystal into liquid nitrogen.

**Data collection and refinement.** Data collection was performed at 100 K using an in-house Rigaku Saturn 92 or Saturn 944+ CCD detector each mounted on a MiraMax-007HF rotating anode generator at a wavelength of 1.54 Å. Data were collected at the Advanced Photon Source (Argonne National Laboratory, Chicago, IL) on the BM22 beamline (Southeast Regional Collaborative Access Team, SER-CAT) using the Mar225 area detector at a wavelength of 1.00 Å. Data were processed and scaled using the program HKL2000[42].

Initial models were determined using molecular replacement with a previously determined structure of pol μ (PDB ID 4M04)[25]. Refinement was carried out using the PHENIX software package[43] and iterative model building was done using Coot[44]. All R$_{free}$ flags were taken from the starting model (PDB ID 4M04); partial catalysis models were generated with both the reactant and product species, and grouped occupancy refinement was performed. Most figures were prepared in PyMol[45], and all omit electron density maps were generated by deleting the regions of interest and performing simulated annealing with harmonic restraints. Ramachandran analysis determined 99.7% of residues lie in allowed regions and at least 97% in favored regions. The outliers (Pro397 and Ser411) are located in loop2 (previously truncated to improve crystallizability[25]), and are on the border of the allowed region of the Ramachandran plot.

**Gap-filling kinetic assays.** Single-turnover kinetic assays were performed on a Kintek RQF-3 chemical quench-flow apparatus (KinTek Corp., Austin, TX), to directly measure the rate of the first insertion ($k_{pol}$) and the apparent equilibrium nucleotide dissociation constant ($K_{d,app}$). Assays were performed in a buffer containing 50 mM Tris-HCl, pH 7.4 (37 °C), 100 mM KCl, 10% glycerol, 100 μg ml$^{-1}$ bovine serum albumin, 1 mM dithiothreitol, and 0.1 mM EDTA. Briefly, 1 μM pol μ was pre-incubated with 100 nM single-nucleotide gapped DNA substrate and rapidly mixed with varying concentrations of metal-TTP (1:1, v:v). The concentration of Mg$^{2+}$ or Mn$^{2+}$ was adjusted to account for equimolar metal binding by TTP, so at least 10 mM Mg$^{2+}$ or 1 mM Mn$^{2+}$ free metal was present. Reactions were quenched with 0.25 M EDTA and mixed with an equal volume of formamide dye. Products were separated on 18% denaturing gels and quantified using a Typhoon phosphorimager and Imagequant software. Time courses were fit to a single exponential, and a secondary plot of the dNTP dependencies on the observed single-exponential rates were fit to a hyperbolic equation[22].

**Data availability.** Coordinates and structure factors for the reported crystal structures have been deposited in the Protein Data Bank under the accession codes 5TXX, 5TXZ, 5TYB, 5TYC, 5TYD, 5TYE, 5TYF, 5TYG, 5TYU, 5TYV, 5TYW, 5TYX, 5TYY, and 5TYZ. Other data that support the findings of this study are available from the corresponding author on reasonable request.

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

## Acknowledgements

The authors thank SER-CAT for assistance with data collection. Use of the Advanced Photon Source was supported by the U.S. Department of Energy, Office of Science, Office of Basic Energy Sciences, under contract W-31-109-Eng-38. This research was supported by research project numbers Z01-ES050158 (S.H.W.), 1ZIA-ES102645 (L.C.P.), and Z01-ES065070 (T.A.K.) in the intramural research program of the National Institutes of Health, NIEHS. We thank Bret D. Freudenthal for discussion during an early stage of this work.

## Author contributions

J.A.J., W.A.B., L.C.P., and S.H.W. designed the project; J.A.J. performed the crystallography. D.D.S. and W.A.B. performed the kinetic experiments; J.A.J., W.A.B., L.C.P., D.D.S., A.F.M., J.M.K. K.B., T.A.K., and S.H.W. analyzed the data; J.A.J., W.A.B., L.C.P., and S.H.W. wrote the paper; S.H.W. directed and supervised the research.

## Additional information

**Competing interests:** The authors declare no competing financial interests.

