## [Peer Review file · Nature Communications]

Reviewers' comments:

Reviewer #1 (Remarks to the Author):

In this manuscript the authors use time-lapsed X-ray crystallography to study the catalytic steps of DNA polymerase mu, an X family polymerase which functions in non-homologous end-joining.

The work presented here include 14 high-resolution crystal structures obtained in Ca²⁺, Mg²⁺ or Mn²⁺, at different time points. Interestingly the structures did not show a transient 3rd catalytic iron, as was described earlier for pol beta and pol eta. Rather, the authors see a transient product metal, but only in the presence of manganese.

The work is of high quality, the figures are clear for the most part and the results are well presented. Pol mu is the third DNA polymerase for which time lapsed X-ray crystallography was used to visualize its catalytic steps. What I personally find quite confusing is that these pol mu crystal structures do not show any evidence of a third metal ion, which was presented as a universal mechanism by the Yang group last year. The product metal is only seen in pol mu, as far as I know. I think it would help the polymerase community tremendously if the authors discussed these differences further.

Lines 311-313: "Surprisingly, and unlike previous observations, the nucleotide Mg²⁺ remains bound to the enzyme after departure of PPi indicating that PPi and nucleotide Mg²⁺ release are uncoupled"

I do not comprehend this statement. What is neutralizing the PPi charge when it is uncoupled from Mg²⁺? What becomes of the coordination of Mg²⁺ in the absence of PPi?

Lines 423-427: "Curiously, the relatively weak binding affinity for dNTPs suggests that pol μ could be limited by nucleotide pool concentrations or utilize NTPs that are often found at higher cellular concentrations.

Have the authors tested whether pol mu can incorporate NTPs?

Figure 1: how were the delta C alpha conformational changes calculated? Which program was used and which part of the proteins were used as reference? In Figure 1, it appears that residues in the palm domain of pol beta undergo substantial movement. For the sake of clarity the authors should label the subdomains of each protein.

As is customary coming from this group the ensemble of structural models is of high quality, based on the crystallographic tables and PDB validation reports. The latter show that at most one of two residues are Ramachandran outliers. Is there something systematic here, i.e, is it almost always the same residues and what is their function?

Reviewer #2 (Remarks to the Author):

The paper by Jamsen et al reports a series of time-resolved X-ray crystal structures of DNA polymerase mu that provide snapshots of the nucleotide-incorporation reaction in the presence of Mg ions and in the presence of Mn ions. This is a very important and interesting study for several reasons.

First, unlike other DNA polymerases for which we have extensive structural information, polymerase mu does not undergo significant domain or sub-domain rearrangements during its catalytic cycle. Its active site is more rigid during catalysis and represents a rather different mechanism of nucleotide incorporation. This paper shows in detail how this active site catalyzes this reaction without large structural changes.

Second, unlike other DNA polymerases, this enzyme is more active in the presence of Mn ions compared to Mg ions. This paper provides not only the structural details of how the Mg-dependent and Mn-dependent mechanisms differ (such as a product metal only appearing in the presence of Mn), it also provides state-of-the-art pre-steady state kinetic analyses of these reactions. (Note that while the interpretation of the thiol effect experiments is controversial in the field, the authors do an outstanding job pointing out the caveats of these experiments.)

Third, there is a great deal of controversy in the literature regarding the role of the third metal ion (the product metal) in the mechanism of DNA polymerases. Time-resolved X-ray crystallographic studies with polymerase beta show that this metal appears after catalysis, and computational studies suggests that the role of the product metal is to inhibit the reverse reaction. By contrast, time-resolved crystallographic studies with polymerase eta have been interpreted as showing that the third metal ion plays a role in facilitating catalysis. The present manuscript supports that interpretation that the product metal ion appears after catalysis.

Thus, overall, this paper makes novel and important contribution to our understanding of polymerase structure and function.

One minor issue is that the referencing of the figures in the text is a little out of order. For example, Fig. 7 is first referenced right after Fig. 3A. As long as this does not violate the journal's stylistic policies, no changes need be made.

Reviewer #3 (Remarks to the Author):

General comments

This manuscript explores the catalytic mechanism of DNA polymerase μ using time lapse x-ray crystallography. The major finding is that in the presence of Mn^{2+} , which is thought to be the relevant co-activator for this polymerase, a third Mn^{2+} appears in the product complex. The third metal doesn't appear when Mg^{2+} is used. Instead, the one of the Mg^{2+} cations is eventually replaced with Na^+ , which causes movement of an aspartate residue that normally interacts with the cation. Included in this study are pre-steady state/single-turnover kinetics using either Mn^{2+} or Mg^{2+} as a cofactor. These are performed with normal TTP and TTP containing a pro-Sp sulfur at the alpha phosphate position. These kinetic experiments highlight the strong preference the enzyme has for Mn^{2+} and suggest that the rate-limiting step of the reaction differs between these two metals (chemistry for Mg^{2+}).

The presence of a third metal interacting with products in the active site of a DNA polymerase is not a new observation, as it has been previously seen in pol β and η (the former being another X-family member). This study on polymerase μ is perhaps the more relevant than pol β , as it is fairly clear that μ prefers Mn^{2+} , although the manuscript does not really uncover the mechanism behind the functional differences between pol μ and β when it comes to metal identity.

Overall, I think the high quality of the structures, the pre-steady state data demonstrating a preference for Mn^{2+} , and the several differences observed between the Mg^{2+} and Mn^{2+} active sites make this manuscript worthy of publication in Nature Communications. This manuscript, when combined with several others, makes it clear that the classic two-metal mechanism for nucleotide incorporation is incomplete.

Specific comments

The authors may want to comment more on the replacement of Mg^{2+} with Na^+ at the 'c' site. Is

this just a product of high Na^+ concentrations in the crystallization solution (a different affinities for Mn^{2+} and Mg^{2+} at this site), or is it relevant to kinetic mechanism? How strong is the evidence for Mg^{2+} being released after each turnover during multiple nucleotide incorporations, or might this be a result of only allowing a single turnover to occur in the crystal?

In line 425, it hasn't been shown that the $\text{Mn}^{2+}(\rho)$ plays a role in slowing the reverse reaction (isn't this just a hypothesis?) – perhaps it should say “Interestingly, a possible role of the product metal in retarding the reverse reaction”.

The scale in Figure 1 is so small (0-1 angstrom) it is hard to tell how large the movements are for pol β . Perhaps different scales could be used for pol μ and pol β ?

Figure 5. Panel C is not mentioned in the figure legend.

Figure 6. It's not really that useful to show just the Mg^{2+} dependent reaction – maybe both metals could be present on the panel B graph (either log-scale y-axis or with Mg^{2+} in an inset?). I think showing the difference between Mn^{2+} and Mg^{2+} would be more useful here.

RESPONSE TO REVIEWERS (NCOMMS-17-00058-T)

General response for the reviewers:

We thank the reviewers for their time and careful reading of the manuscript.

REVIEWER #1

General Comments.

"... What I personally find quite confusing is that these pol mu crystal structures do not show any evidence of a third metal ion, which was presented as a universal mechanism by the Yang group last year. The product metal is only seen in pol mu, as far as I know. I think it would help the polymerase community tremendously if the authors discussed these differences further."

Response: What the Yang group refers to as the "third metal", we refer to as the "product metal", since it is only observed after nucleotide insertion, and coordinates the products of the reaction (i.e., the inserted dNMP and PPI). While the reviewer correctly points out that the Yang group proposed that the "three-metal" mechanism is universal, the observation that pol mu does not include a third (product) metal with Mg²⁺ indicates that the product metal is not essential or directly involved in nucleotide insertion chemistry. Computational studies with pol beta suggest that the product metal does not have a role in the forward nucleotidyl transfer reaction (Perera et al., 2017; ref. 12), but deters the reverse, pyrophosphorolysis, reaction (Perera et al., 2015; ref. 13) (lines 42-43 of the revised manuscript). Additional evidence against involvement of a product metal ion in catalysis is that a lysine side chain in A- and B-family DNA polymerases occludes the third-metal binding site. In order to clarify that the third and product metal are the same metal, we refer to this adjunct metal in the revised manuscript as the "third (product) metal". Additionally, we highlight the fact that a third or product magnesium is not observed with pol mu, demonstrating that the three-metal mechanism is not universal (line 306 of the revised manuscript).

Specific Comments (SC).

SC (1) Lines 311-313 (lines 312-314 of the revised manuscript): "Surprisingly, and unlike previous observations, the nucleotide Mg²⁺ remains bound to the enzyme after departure of PPI indicating that PPI and nucleotide Mg²⁺ release are uncoupled"

I do not comprehend this statement. What is neutralizing the PPI charge when it is uncoupled from Mg²⁺? What becomes of the coordination of Mg²⁺ in the absence of PPI?

Response: Since water penetrates the active site, water now coordinates the nucleotide metal. Subsequent to catalysis, product dissociation should be encouraged to permit cycling. Accordingly, the increased negative charge on PPI lacking a bound metal would promote dissociation. The nucleotide metal would most likely dissociate concomitant with DNA translocation, since it would lose a coordinating ligand (phosphate oxygen on the nascent nucleotide).

SC (2) Lines 422-427: "Curiously, the relatively weak binding affinity for dNTPs suggests that pol mu could be limited by nucleotide pool concentrations or utilize NTPs that are often found at higher cellular concentrations.

Have the authors tested whether pol mu can incorporate NTPs?

Response: Previous work has demonstrated that pol mu exhibits poor sugar discrimination and readily inserts ribonucleotides (Nick McElhinny et al., 2003; ref. 30). These structural studies are currently underway.

SC (3) Figure 1: how were the delta C alpha conformational changes calculated? Which program was used and which part of the proteins were used as reference? In Figure 1, it appears that residues in the palm domain of pol beta undergo substantial movement.

For the sake of clarity the authors should label the subdomains of each protein.

Response: The Matchmaker tool in the Chimera package (UCSF) (Pettersen et al., 2004; ref. 46) was used to align the protein alpha-carbons of the binary DNA and ternary (DNA/dNTP) complexes, as well as map alpha-carbon displacements onto a ribbon representation of the ternary complex. All alpha-carbons (i.e., 325 residues) of the respective pol mu structures were used to perform the alignment since it does not display subdomain motions (RMSD 0.23 angstrom). In contrast, since the N-subdomain (fingers of right-handed DNA polymerases) of pol beta undergoes repositioning when a nucleotide binds to form the ternary complex, the alignment was restricted to the other polymerase beta subdomains (RMSD 0.90 angstrom, 224 residues).

Figure 1 displays a limited range of alpha-carbon displacements (0-1 angstrom) to highlight the rigidity of the protein backbone of pol mu. Thus, the apparent mobility of the pol beta palm (i.e., C-subdomain) is limited, whereas the N-subdomain can move nearly 10 angstroms (cf., Figure 4A in Beard and Wilson, 2014; ref. 20). We have added a detailed description of the alignment in the figure legend (Lines 650-658 of the revised manuscript), and have updated Figure 1 with subdomains labeled. Since the 8-kDa domain of pol mu does not have lyase activity, it is simply referred to as the 8-kDa domain, whereas it is referred to as the lyase (L) domain with pol beta.

SC (4) As is customary coming from this group the ensemble of structural models is of high quality, based on the crystallographic tables and PDB validation reports. The latter show that at most one of two residues are Ramachandran outliers. Is there something systematic here, i.e, is it almost always the same residues and what is their function?

Response: The outlier present in one of the structures was eliminated. The remaining outliers, present in some structures (Pro397 and Ser411), are part of loop2 that was truncated (Pro398-Pro410) to improve the crystallizability of the pol mu construct (see Moon et al., 2014; ref. 25). The outliers are on the border of the allowed region of the Ramachandran plot, are distant from the active site, and do not appear to be involved in catalysis. Reports on the function of these residues are not available in the literature. See lines 475-479.

REVIEWER #2

SC (1) One minor issue is that the referencing of the figures in the text is a little out of order. For example, Fig. 7 is first referenced right after Fig. 3A. As long as this does not violate the journal's stylistic policies, no changes need be made.

Response: We have corrected this; Figure 7 is now Figure 4.

REVIEWER #3

SC (1) The authors may want to comment more on the replacement of Mg^{2+} with Na^{+} at the 'c' site. Is this just a product of high Na^{+} concentrations in the crystallization solution (a different affinities for Mn^{2+} and Mg^{2+} at this site), or is it relevant to kinetic mechanism? How strong is the evidence for Mg^{2+} being released after each turnover during multiple nucleotide incorporations, or might this be a result of only allowing a single turnover to occur in the crystal?

Response: The loss of Mg^{2+} from the catalytic site after insertion most likely reflects the lower affinity of the catalytic site for this metal when a coordinating ligand is lost (i.e., primer $O3'$) (line 340 of the revised manuscript). This binding site will not be regenerated until the polymerase translocates one nucleotide moving the primer terminus to the boundary of the dNTP binding pocket. The high sodium concentration in the crystallization solution is probably partly responsible for its occupation of the catalytic site. Unpublished data with pol beta indicates that potassium, the physiological monovalent cation, is too large to occupy this site. The ability of Mn^{2+} to form altered coordination spheres suggests that it could bind to a suboptimal Mg^{2+} binding site. Since the catalytic site must be filled with Mg^{2+} for a reverse reaction to occur (Perera et al., 2015; ref. 13), it is thermodynamically advantageous not to fill this site to drive the reaction forward. Likewise, the metal in the nucleotide metal site would be expected to dissociate when DNA translocates, since its coordinating ligand is lost (phosphate oxygen on the nascent nucleotide). See lines 358-366.

SC (2) In line 425, it hasn't been shown that the $Mn^{2+}(p)$ plays a role in slowing the reverse reaction (isn't this just a hypothesis?) – perhaps it should say “Interestingly, a possible role of the product metal in retarding the reverse reaction”.

Response: This statement should be accompanied with a reference to a computational study that indicated that this metal deters the reverse reaction (Perera et al., 2015; ref. 13). Additionally, recent computational studies of the forward reaction indicate that the third metal does not alter the reaction barrier (Perera et al., 2017; ref. 12). See lines 358-360 of the revised manuscript.

SC (3) The scale in Figure 1 is so small (0-1 angstrom) it is hard to tell how large the movements are for pol beta. Perhaps different scales could be used for pol mu and pol beta?

Response: The scale was limited to highlight the rigidity of the pol mu backbone. We have added a reference, in the legend of Figure 1, that includes a similar figure for the backbone displacement (0-10 angstrom) exhibited by pol beta (Fig. 4A in Beard and Wilson, 2014; ref. 20).

SC (4) Figure 5. Panel C is not mentioned in the figure legend.

Response: This has been corrected. The structure is the product state (PS) after the extended soak, showing that while the product metal and PPI have dissociated, while the catalytic and nucleotide metals remain bound. Figure 5 is now Figure 6 in the revised manuscript.

SC (5) Figure 6. It's not really that useful to show just the Mg²⁺ dependent reaction – maybe both metals could be present on the panel B graph (either log-scale y-axis or with Mg²⁺ in an inset?). I think showing the difference between Mn²⁺ and Mg²⁺ would be more useful here.

Response: We have included the Mn²⁺ data in the figure (adjacent to the Mg²⁺ data). Inspection of the time scales of the plots highlight the difference in the insertion rates for the two metals (i.e., Mg²⁺, 0–10 s; Mn²⁺, 0–0.4 s). Note that Figure 6 is now Figure 7 in the revised manuscript.